# A Procedure for Complete Census Estimation of Rooftop Photovoltaic Potential in Urban Areas

**Andrea Pinna and Luca Massidda \***

CRS4, Center for Advanced Studies, Research and Development in Sardinia, loc. Piscina Manna ed. 1, 09050 Pula (CA), Italy; andrea.pinna@crs4.it

**\*** Correspondence: luca.massidda@crs4.it

**Abstract:** Rooftop photovoltaic solar systems can be an essential tool to support the energy transition of Europe. The assessment of solar power generation potential in urban areas, necessary for smart grid planning, requires the processing of data of different types, such as building cadastral information, a detailed description of available roof areas, and solar irradiation data. We introduce an algorithm for the fast calculation of the building's shadows and a procedure for the integration of solar irradiation in time. We therefore develop a methodology that allows a fast evaluation with minimal computational resources, and we apply it to an urban scenario of a medium-sized European city obtaining an estimate of the complete census PV power generation potential, with a spatial resolution of 1 m. We validate the results by comparison with a reference procedure, obtaining minimal deviation with a much lower demand for computational resources.

**Keywords:** rooftop solar photovoltaic systems; photovoltaics; potential estimation; urban solar planning

## 1. Introduction

The targets for the reduction of greenhouse gas (GHG) emissions set out in the Paris Agreement of 2015 determine the need for a strong decarbonisation of energy supply systems, in this context great attention is paid to distributed generation with the use of roof-mounted photovoltaic panels [1,2]. The International Energy Agency forecasts a large increase in the installation of photovoltaic systems in urban areas, estimating that the residential photovoltaic market could triple its volume by 2030. In the European Union, since the publication in 2009 of the first European Renewable Energy Directive on grid-connected photovoltaic systems, the installed power has increased tenfold, from 11.3 GW at the end of 2008 to over 116 GW at the end of 2018 [3]. The recent (2018) revised European Renewable Energy Directive has set a target of 32% as a fraction of energy consumption from renewable sources by 2030 [4]. This is certainly an ambitious target, which will require a significant increase in power generation from renewable sources (at least 65%). Part of it will have to come from photovoltaic generation [5]. Almost half of the photovoltaic production capacity in Europe comes from rooftop, residential (28%) or commercial (18%) installations. Intensifying the diffusion of photovoltaic modules would facilitate the achievement of the above objectives. A similar development naturally leads one to wonder what the actual potential for photovoltaic production for residential and commercial buildings in the European Community could be, and whether this potential could be compatible with the objectives set.

Following Castellanos et al. [6] it is possible to divide the methods for the evaluation of photovoltaic potential in urban areas into three levels called *low*, *medium* and *high*. The *low level* methods, based on a relationship between population density and the type of construction and roof area available for PV installation, showed low reliability; *medium level* methods associate statistical information with spatial information derived from Geographical Information System (GIS) and Light

Detection And Ranging (LiDAR)-based methods; finally, the *high level* methods use high resolution geographical spatial information about the actual solar radiation, and often include, in addition to the actual rooftop surfaces, their inclination and orientation, and the effects of shading.

According to the classification proposed by Byrne et al. [7], the methodologies for the evaluation of potential are divided on the basis of the employed data: (i) *sampling-based methodologies*, in which the potential for a sample area is accurately assessed and the result is extrapolated to the surrounding areas; (ii) *multivariate sampling-based methodologies*, in which correlations are identified between statistical data, such as population density and type of buildings, and the availability of suitable roof area; (iii) *complete census methodologies*, in which all available high resolution geographical information on the examined area is used, without extrapolation from sampling.

Such assessments in metropolitan areas have already been carried out in the literature, thanks to the progressive growth in computational resources and the availability of increasingly detailed geographical data. Nguyen et al. outlined a LiDAR-based procedure to identify buildings suitable for panel installation and applied it to a very small area of about 50 buildings in the city of Kingston, ON (Canada) [8]. Two other similar LiDAR-based techniques have been applied over Knox County, TN (United States) by Kodysh et al. [9] and over the city of Georgetown in Malaysia by Latif et al. [10]. Through the LiDAR reconstruction it is also possible to take into account the slope of the roof surfaces when estimating the generation potential. The availability of satellite imagery and the development of automatic image segmentation techniques allowed Khan et al. to identify the available surfaces for a district of Karachi in Pakistan, and to estimate the production potential based on monthly averages of irradiance data [11].

The increased availability of computational power and high resolution geographical data has allowed for analyses to be carried out on larger metropolitan areas, such as the Gangnam district of the city of Seoul in South Korea [12]. Hong et al. derived building data from a high resolution GIS database and through coupling with hourly irradiance data, assessed the potential for photovoltaic generation, by taking also into account the shading of buildings. The potential of the entire city of Seoul was already been assessed in [7], but without taking shading into account and with an estimate of the usable rooftop area obtained on a statistical basis and verified by cartographic analysis. The estimate for the city of Mumbai in India is instead conducted by Singh et al. with a *multivariate sampling-based* methodology [13]. Another technique of the same class, based on sample analysis and extrapolation from 128 major cities in the United States, was presented by Margolis et al. [14]. Bodis et al. presented an assessment of the potential for the entire European territory, based on a *complete census* methodology where a 100 m resolution grid has been analyzed [5].

The studies described above and the vast majority of tools available for the precise evaluation of the energy potentially produced by a photovoltaic system are based on the estimation of the radiation on the Plane Of Array (POA) from a combination of the direct and diffuse components of solar radiation and from the panel orientation. These evaluations allow an assessment of the production over a typical year, using data available from public services such as PVGIS (Photovoltaic Geographical Information System) [15] or NREL (National Renewable Energy Laboratory) [16]. However, given the amount of calculations required for the application of an irradiation-based procedure for very large domains, as is the case for the whole territory of a city, approximate methods have been proposed which only aim at a full year assessment of the energy that can be produced.

The Sky View Factor (SVF), used as an irradiation indicator, i.e., a function of the obstruction of the horizon by surrounding buildings on an exposed surface, is introduced by Robinson et al. [17]. Moreover, with the aim of reducing the time requested by calculations, Rodriguez et al. proposed another methodology based on a *clear sky* model and a three-dimensional representation of buildings [18].

More recently, Calcabrini et al. introduced an approximate methodology for the assessment of the production potential even in the presence of shading due to complex urban *skylines*, based on an elaboration of the SVF and validated through actual production data of some PV plants [19].

Walch et al. proposed a study on the producibility of the entire territory of Switzerland, also addressing the problem of estimating the temporal variability of production [20]. Finally, Tiwari et al. propose a procedure for the estimation of the rooftop solar energy photovoltaic potential using LiDAR scans and ortho-rectified aerial photography, without the need to use cadastral data, and apply it to a small city in Israel [21].

In this work we want to propose a *high level* and *complete census* methodology for a fast and accurate evaluation of rooftop photovoltaic production potential by mapping the radiation time series according to the apparent position of the Sun, and by quickly calculating the shading in urban areas from building elevation data. We apply the proposed procedure to a real domain and validate it by comparison with a reference procedure.

The article is structured as follows: Section 2 presents the adopted methodologies and Section 3 describes the application of the proposed approach to the city of Cagliari in Italy, while the results are presented in Section 4 and discussed in Section 5. Finally, Section 6 presents the conclusions of this work and its possible developments.

## 2. Methodology

### 2.1. Introduction

To carry out a complete census of the radiation available for photovoltaic generation from rooftop panels in an urban environment, it is necessary to have (i) a description of the surfaces potentially available for the installation of panels (Section 2.2), (ii) an accurate estimate of the incident solar radiation in a typical year for an exposed surface with no irradiation obstacles (Section 2.3) and (iii) a precise estimation of shading, which may reduce the actual radiation on such surface (Section 2.4).

### 2.2. Surface Availability

The area actually available for the installation of a photovoltaic system in a building depends on the shape of the roof itself, on its slope and orientation and on the presence of technical superstructures that prevent installation. The exact evaluation of such surface would require a precise analysis of the characteristics of the individual buildings and is beyond the scope of this work; however, it is possible to obtain an estimate by using a high resolution Digital Surface Model (DSM) of the urban area considered.

Various methods have been proposed in the literature to estimate the effective area available for the installation of photovoltaic systems. Generally these methods provide the calculation of a coefficient that describes the ratio of usable roof surface with respect to the total, depending on the type of building and its architectural characteristics [20]. In our case we propose a criterion exclusively based on the analysis of the building elevation data. The procedure is summarized in three main steps: (i) calculation of the orientation and slope of the building rooftops (such quantities will be later also employed for the evaluation of the photovoltaic potential of the panels) using a high resolution DSM; (ii) removal of the portions of building rooftops characterized by a slope greater than $45°$, which we assume denote the presence of obstacles or discontinuities; (iii) selection, among the remaining rooftop surfaces, of all the contiguous areas of at least 30 m$^2$.

The limit of 30 m$^2$ corresponds to an area for which a system of at least 3 kW$_P$ could be installed, below which the installation of a system for the radiation conditions of the city under examination does not appear justified. The $45°$ threshold, on the other hand, is used to identify the presence of obstacles or raised structures; the roofs of the area under examination are in fact flat or with moderate slope. The limits are similar to those resulting in the analysis in [21].

*2.3. Irradiation Computation*

The annual energy yield of a photovoltaic module can be obtained by estimating the environmental conditions and the irradiation on its surface, and by defining the characteristics of the panel itself.

The solar radiation reaching the earth's surface is commonly divided in two distinct components: the first component, known as Direct Normal Irradiation (DNI), is associated with energy coming directly from the Sun following the path of its rays; the second component reaches the surface indirectly, and derives from the portion of radiation diffused from the sky. The ground projection of the diffused radiation is indicated as Diffuse Horizontal Irradiation (DHI). The sum of direct and diffuse irradiation projected to the ground yields the Global Horizontal Irradiation (GHI) [22]:

$$\text{GHI} = \text{DHI} + \text{DNI} \cdot \cos\theta, \tag{1}$$

where $\theta$ is the solar zenith angle.

Following the approach already used in [23,24] we break down the radiation that reaches a non horizontal surface into the sum of the projections of the direct and diffuse components of the solar radiation, net of any obstacles and shading. The radiation on the (POA) is calculated as [25]:

$$E_{\text{POA}} = E_b + E_g + E_d, \tag{2}$$

where $E_b$ indicates the beam component, $E_g$ indicates the component diffused from the ground and $E_d$ indicates the component diffused from the sky. The solar radiation beam component $E_b$ corresponds to the DNI projection on the surface [25]:

$$E_b = \text{DNI} \cdot \cos(\text{AOI}), \tag{3}$$

where AOI (Angle Of Incidence) denotes the angle of incidence between the Sun rays and the surface. The sky diffusion component $E_d$ is calculated from the DHI and the panel tilt. Several models are available for its estimation, among which the simplest is the isotropic model [25]:

$$E_d = \text{DHI}\frac{1 + \cos\beta}{2}, \tag{4}$$

in which $\beta$ represents the slope of the panels compared to the horizontal plane.

Finally, the component diffused by the terrain $E_g$ can be calculated as [26]:

$$E_g = \rho \cdot \text{GHI}\frac{1 - \cos\beta}{2}, \tag{5}$$

where in the urban environment [27], the albedo $\rho = 0.18$.

With the ground irradiation time series data available, the total irradiance on the plane of the photovoltaic module is calculated using Equation (2) for all instants in the time series.

The resulting values, together with the air temperature and wind conditions, may then be processed through a physical model that makes it possible to determine the electrical power generated at any given time, such as the PVWATTS model [16] or the model implemented in PVGIS [28]. All these models require the definition of one or more calibration parameters, characteristic of a given technology and a specific panel supplier. In consideration of the rapid evolution of the technology, and consequently of the conversion efficiency achievable from the panels, we decided in this work to focus on the estimation of the effective radiation on the roof surface. The calculation of the achievable energy will depend only on the conversion efficiency, in turn function of the chosen and available technology.

## 2.4. Shadowing Computation

In an urban environment the presence of surrounding buildings can reduce the useful radiation reaching a photovoltaic panel and should thus be taken into account.

Therefore, Equation (2) is modified to account for the possible shading:

$$E_{\text{POA}} = F_s(\theta, \alpha, P) \cdot E_b + E_g + E_d, \qquad F_s \in \{0, 1\} \tag{6}$$

where $F_s$, a function of the position of the point $P$ and the apparent position of the Sun through the angles of zenith $\theta$ and azimuth $\alpha$, has unitary value if the Sun is directly visible to an observer on the panel centered on the point $P$, and null value if it is hidden by the apparent horizon. The procedure for calculating the energy obtainable from the panel remains otherwise unchanged.

A widely used approach in the literature for the calculation of shading due to other buildings is to evaluate for each individual point of the domain or for each building the height of the apparent horizon [20,29]. This is the standard procedure used by PVGIS [15], a reference tool for the calculation of photovoltaic production potential. However, the application of this procedure to the entire available rooftop area in an urban scenario is particularly demanding when applied with a high spatial resolution, mainly due to the time needed for the GIS system to calculate the horizons.

We therefore introduce an alternative algorithm to calculate the shadows projected by a light source on the points within an area described by a DSM, thus obtaining an approximation of the $F_s$ function from Equation (6) for each apparent position of the Sun. The software is open-source and available at https://github.com/pinno/shadow-mapper.

The algorithm, illustrated in Figure 1, consists of a variation of the classic Bresenham algorithm [30] used in computer graphics for the drawing of lines: given a point $P$ of the DSM raster map which belongs to the rooftop of a building, we consider the half-line that starts from the point $P$ and is directed to the position of the Sun $S$; a generic point on this half-line is indicated with $R$. The points $R'$ belonging to the projection on the ground of the ray $(P, S)$ can be quickly calculated with the Bresenham algorithm; $Q$ is the point of the DSM whose projection to the ground corresponds to the point $R'$. If for any of the points $Q$ the elevation is higher than the elevation of the corresponding points $R$, then the point $P$ is in shadow. Iterating over the points $P$ yields the map of the shadows projected from the Sun to all the building rooftops for a given position of the Sun.

Algorithm 1 is formulated through two nested cycles: the external cycle describes all the points on the raster map belonging to the surface of the building rooftops, while the internal cycle describes the position of point $R$ along the ray $(P, S)$ and it exits when the previously described condition of shadow occurs, when the projection $R'$ of the point $R$ reaches the limit of the domain, or when the height of the point $R$ exceeds the maximum height of the DSM. This allows for assembling a shadow map for each instant of a typical year, i.e., which parts of a building rooftop are in shade for any given pair of angles $(\alpha, \theta)$ describing the apparent position of the Sun.

It would therefore be necessary to repeat the calculation of the shadow map for each instant of the irradiation time series, in order to obtain an estimate of the annual production of a photovoltaic panel. The procedure can be optimized by subdividing the positions of the Sun in the sky into sectors, following an approach similar to that proposed in [31] for the calculation of the Sky View Factor.

---

**Algorithm 1:** Shadow map algorithm

---

**Input:** $H[i, j]$: A float 2D array ($n_i$ rows and $n_j$ columns) of the heights of the DSM
$P'i[k], P'j[k]$: Two integer 1D arrays of length $n_k$ containing the coordinates (row $i$ and column $j$) of the pixels corresponding to the building rooftops on the DSM
$\Delta$: The spatial resolution of the DSM
$\alpha, \theta$: The Sun position: azimuth, zenith
**Output:** $Fs[k]$: A binary 1D array of length $n_k$ indicating if the rooftop cell $k$ in the map is lit
($Fs[k] = 1$) or in shade ($Fs[k] = 0$)
$s_x = \sin(\alpha)\sin(\theta)$ ;
$s_y = -\cos(\alpha)\sin(\theta)$ ;
$s_z = \cos(\theta)$ ;
**for** $k \leftarrow 0$ **to** $n_k - 1$ **do**
    $Fs[k] = 1$ ;
    $x = (\text{float})P'i[k]\Delta$ ;
    $y = (\text{float})P'j[k]\Delta$ ;
    $z = H[P'i[k], P'j[k]]$ ;
    **while** *($x \geq 0$) and ($x < n_i\Delta$) and ($y \geq 0$) and ($y < n_j\Delta$) and ($z \leq max(H)$)* **do**
        **if** *($z < H[(int)x, (int)y]$)* **then**
            $F_s[k] = 0$ ;
            **break**
        $x = x + s_x$ ;
        $y = y + s_y$ ;
        $z = z + s_z$ ;

---

The total annual irradiance over a horizontal area $E^{\text{GHI}}$ can be calculated from the time series of the global irradiance GHI as an integral over time, and approximated by a sum of the series values when the sampling interval $\Delta t$ is constant.

$$E^{\text{GHI}} = \int_0^T \text{GHI}(t)dt \approx \Delta t \sum_{i=0}^N \text{GHI}_i \tag{7}$$

$$\text{GHI}_i = \text{GHI}(t_i) = \text{GHI}(\alpha(t_i), \theta(t_i)) = \text{GHI}(\alpha_i, \theta_i) \tag{8}$$

The terms $\text{GHI}_i$ can be expressed as a function of the azimuth ($\alpha$) and zenith ($\theta$) angles of the Sun apparent position. By choosing a resolution $\delta$ for the angles above, the latter can be approximately represented as $\alpha_i = p_i\delta$ and $\theta_i = q_i\delta$. Therefore, Equation (7) becomes:

$$E^{\text{GHI}} \approx \Delta t \sum_{p,q} E_{p,q}^{\text{GHI}} \tag{9}$$

where the Iverson bracket is adopted:

$$E_{p,q}^{\text{GHI}} = \sum_{\alpha_i, \theta_i} [\alpha_i = p\delta][\theta_i = q\delta]\text{GHI}(\alpha_i, \theta_i) \tag{10}$$

$$E_{p,q}^{\text{DNI}} = \sum_{\alpha_i, \theta_i} [\alpha_i = p\delta][\theta_i = q\delta]\text{DNI}(\alpha_i, \theta_i) \tag{11}$$

$$E_{p,q}^{\text{DHI}} = \sum_{\alpha_i, \theta_i} [\alpha_i = p\delta][\theta_i = q\delta]\text{DHI}(\alpha_i, \theta_i) \tag{12}$$

This reformulation allows for a much faster calculation of the radiation on a surface even in the presence of shading: since shadows only affect the direct radiation, the calculation of the $E^{\text{DNI}}$

terms will have to be done not for all the instants of the time series, but just for the $p, q$ index pairs for which the value of $E_{p,q}^{\mathrm{DNI}}$ is not null (Equation (11)). A similar reformulation applies to the diffuse DHI component, as expressed in Equation (12). The DHI component is not influenced by the building shadows.

The value of $\delta$ determines the precision of the calculation and will be a function of (i) the available computing power, (ii) the resolution of the DSM and (iii) the desired precision. Consequently, the average irradiance on a oriented surface is calculated by applying Equations (3), (5) and (6) for each sector of the sky and then by adding up the resulting values.

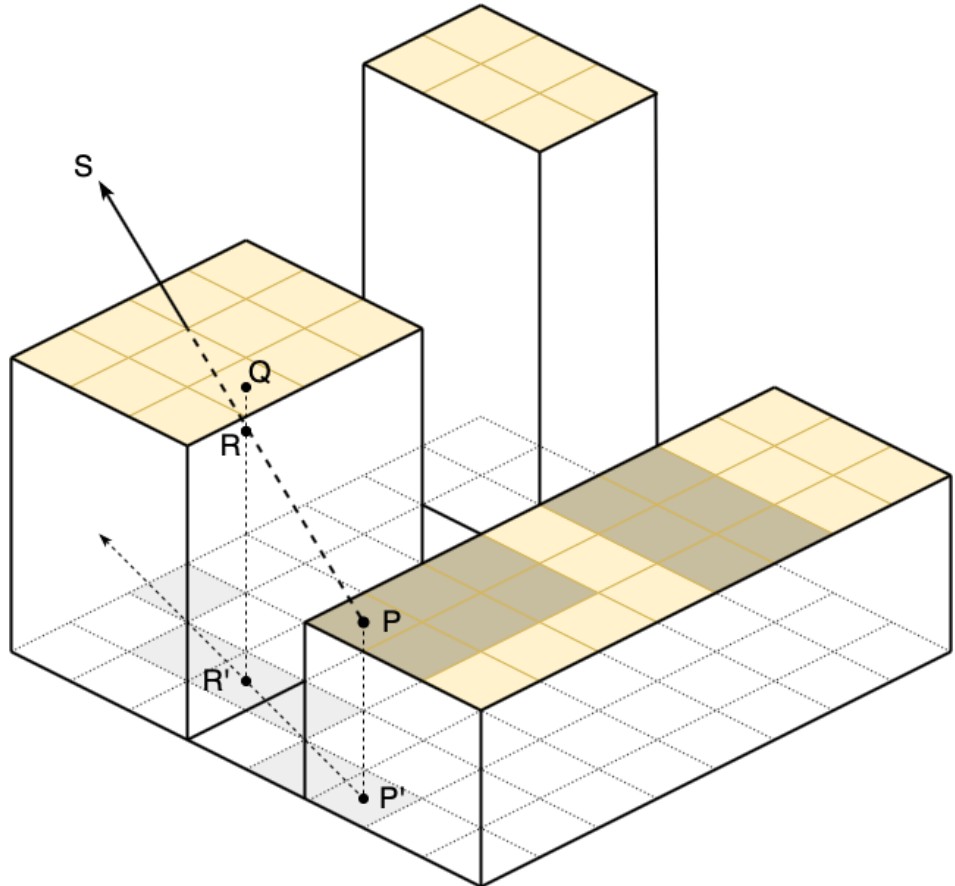

**Figure 1.** Algorithm for the calculation of the shading map. The point $P$ moves on the DSM surface of the buildings of the considered domain, the $R$ point moves from the $P$ point in the direction of the Sun along a ray, the $P$ and $R'$ points represent the projections of $P$ and $R$ at zero elevation, and the ground projection of the DSM point $Q$ coincides with point $R'$.

## 3. Method Application

### 3.1. Introduction

The application of the proposed methodology for the identification of rooftops suitable for photovoltaic systems requires the availability of precise and high resolution data for the meteorological variables that determine solar radiation, for the elevation model that describes the volumetric occupation of the area studied and for the cadastral footprints and attributes that permit the selection of suitable buildings.

This study focuses on the metropolitan area of Cagliari, the capital of Sardinia, one of the 20 regions of Italy. The city, which has a population of about 150,000 inhabitants, is situated on a gulf and enjoys a mild climate with low rainfall. It is also characterized by the presence of several hills that, in the earliest and late hours of the day, cast shadows on the buildings below. For some years

now the city of Cagliari has also been the subject of study in the field of Smart Cities, such as the project Tessuto Digitale Metropolitano (http://www.tdm-project.it) and the project SARDINE (http://www.crs4.it/projectdetails/18bb2f43-bfd5-11e8-ae20-506b8da9258c/), in whose context this work is inserted.

## 3.2. Meteorological Data

Historical hourly solar radiation time series are obtained from the PVGIS web platform, developed by the Joint Research Centre (JRC) of the European Commission. The PVGIS-CMSAF (Satellite Application Facility on Climate Monitoring) and PVGIS-SARAH (Solar surfAce RAdiation Heliosat) datasets are available for the European territory, both calculated from satellite data and validated on several ground stations [15,32–35]. Satellite data is preferred over ground measurements because it provides greater coverage with higher resolution. The time series of the CMSAF dataset are available for a period of 10 years (2007 to 2016, included), and those of the SARAH dataset for 12 years (2005 to 2016). The temporal resolution is hourly, while the spatial resolution, at Cagliari latitude, is about 2.5 km for the CMSAF dataset and about 5 km for the SARAH dataset. The historical time series of direct, diffuse and ground reflected radiation on a horizontal surface, the average air temperature and the wind speed 10 m above ground are available for download. It should be noted that the CMSAF dataset will be removed from PVGIS as it is characterized by significantly more uncertainty about the measurements than the SARAH dataset. The PVGIS system, accessible both through the graphical user interface (https://ec.europa.eu/jrc/en/pvgis) and through API queries (https://ec.europa.eu/jrc/en/PVGIS/docs/noninteractive), allows not only to obtain historical irradiation data for a typical year, but also to calculate the irradiation on a specific panel and therefore the time series of the power generated by an installation for which the technological characteristics are provided. The system also allows taking into account the height of the horizon in order to cancel the direct component of the radiation for the period of time during which the elevation of the Sun is lower than such height.

## 3.3. Digital Surface Model

The Sardegna Mappe service offered by Regione Sardegna (http://www.sardegnageoportale.it/webgis2/sardegnamappe/?map=download_raster) provides the Digital Terrain Model (DTM) and Digital Surface Model (DSM) built from the point cloud detected by a LiDAR Laser Scanning System and released with an accuracy of 1 m. The DSM gives the elevation from sea level for each point of the 1 m resolution grid, also including building and vegetation encumbrance; this allows calculating the shading within the municipal boundaries when the apparent position of the Sun changes. In addition, an estimate of the inclination and orientation of surfaces can be obtained from the DSM itself by applying the Horn [36] algorithms, available through popular GIS software libraries, e.g., [37]. Figure 2 shows the city boundaries and a terrain representation of Cagliari heights from the adopted DSM.

## 3.4. Building Data

The cadastral information is taken from the Land Information System (Sistema Informativo Territoriale, SIT) established by the municipality of Cagliari (https://sit.comune.cagliari.it), from which the city perimeter can be derived, and from the GeoTopographic DataBase (DataBase GeoTopografico, DBGT) instituted by the regional government of Sardinia (http://www.sardegnageoportale.it/areetematiche/databasegeotopografico), which provides various GIS data such as information on the building's use and the coordinates of the perimeter vertices, including height above ground. Buildings that are not suitable for photovoltaic panel installations (e.g., monuments, churches, etc.) will therefore not be considered in the procedure presented. All of the above data enables the accurate estimate of the horizontal surface potentially available for the installation of photovoltaic panels. Figure 2 shows the building perimeters obtained from the DBGT dataset for a portion (highlighted by the red square) of the examined domain.

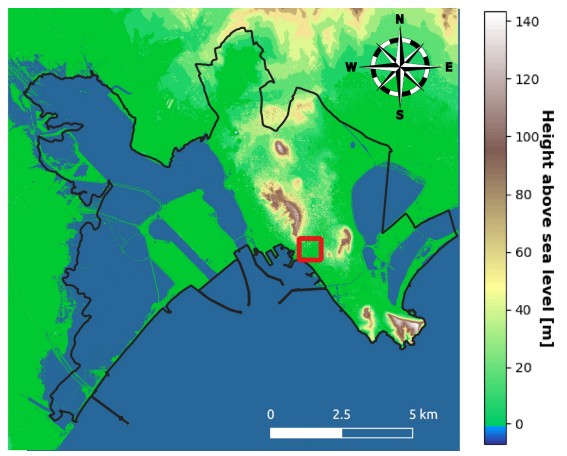
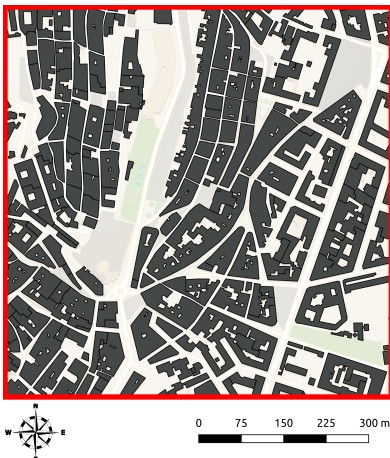

(**a**) DSM and boundaries of the city of Cagliari. (**b**) Building footprints in downtown Cagliari.

**Figure 2.** The digital model of the city surface and the surrounding areas: the territory of Cagliari is within the black line; the buildings are mainly concentrated in the middle-eastern sector, while the western one is almost entirely occupied by the Santa Gilla lagoon, the salt pans and the Porto Canale commercial harbour. The area shown in (**b**) is positioned within the red square shown in (**a**).

## 3.5. Proposed Methodology

Thanks to the available data, listed in the previous section, it is possible to estimate the irradiation on the roofs of the buildings of the city of Cagliari, and consequently the production of photovoltaic energy from panels hypothetically placed on the roofs themselves, by (i) obtaining the time series of irradiation measurements from the datasets offered by PVGIS, (ii) calculating the shadow map according to the angles of zenith and azimuth for the desired resolution and (iii) estimating the irradiation on the useful surfaces of the buildings.

The proposed approach, reported in Algorithm 2, allows obtaining a detailed map of the direct and diffuse radiation on all the roof surfaces of the buildings within the domain. Then, by also employing the assessment of the rooftop slope and orientation, it is finally possible to obtain an estimate of the irradiation on the photovoltaic panels.

---

**Algorithm 2:** Calculation of irradiance at the tops of buildings.

**Input:** PVGIS dataset: selection between PVGIS-CMSAF and PVGIS-SARAH;
$\beta_{\max}$: maximum allowed surface inclination;
$S_{\min}$: minimum contiguous surface area requested for the installation of a photovoltaic system;
$\delta$: angle discretization $\alpha$ and $\theta$;
**Output:** $E_{\mathrm{POA}}$ for all cells $R_{ijk}$ into which the rooftops of the buildings within the city of
        Cagliari are subdivided;
· subdivision of the territory of Cagliari into macro-areas $A_i \in \mathcal{A}$ according to the resolution of
  the chosen dataset: the irradiation calculated by PVGIS is in fact constant, for every temporal
  instant, for all the points located within the macro-area $A_i$;
· calculation of the inclination $\beta$ and orientation $\gamma$ for each point of the territory according to
  the resolution of the DSM raster;
**for** $A_i \in \mathcal{A}$ **do**
    · requesting and downloading from PVGIS the time series of GHI, DNI and DHI of the
      macro-area $A_i$;
    · grouping and integral sum of GHI, DNI and DHI values according to the $\alpha$ and $\theta$ angles
      of the apparent position of the Sun according to Equations (10)–(12);
    · identification of buildings $E_{ij}$ located within $A_i$;
    **for** $E_{ij} \in \mathcal{E}_i$ **do**
        · subdivision of the building's surface area $E_{ij}$ in cells $R_{ijk} \in \mathcal{R}_{ij}$ side equal to 1 m, thus
          corresponding to the resolution of the DSM;
        · removal from the $\mathcal{R}_{ij}$ set of the cells $R_{ijk}$ with a slope above the angle $\beta_{\max}$;
        · removal from the $\mathcal{R}_{ij}$ set of the cells $R_{ijk}$ belonging to contiguous surfaces with an
          extension of less than $S_{\min}$;
        **for** $R_{ijk} \in \mathcal{R}_{ij}$ **do**
            · calculation of the shading factor $F_s$ using Algorithm 1 for each pair $(\alpha_p, \theta_q)$
              describing the apparent position of the Sun;
            · calculation of the effective irradiance map $E_{\mathrm{POA}}$ according to Equations (3)–(6),
              then taking into account the shading factor $F_s$, the slope $\beta$ and the orientation $\gamma$ of
              the cell $R_{ijk}$.

---

### 3.6. Benchmark Methodology

To verify the validity of the proposed approach, the estimation of the average annual overall irradiance on the horizontal surface of rooftops was compared with the same quantity calculated by well known in the literature and publicly available GIS tools. We therefore used a combination of GRASS GIS and PVGIS for obtaining the estimate of the average annual irradiance measured on the the selected buildings. In order evaluate the shading for the direct normal irradiance, the apparent horizon due to surrounding buildings and natural obstacles has been computed, for each cell of 1 m$^2$ contained within the rooftops perimeter, through the module `r.horizon` of GRASS GIS [38,39] . The horizon height has been calculated with an angular resolution of 2° for the azimuth and defines the minimum elevation angle beyond which the Sun is visible, i.e., the threshold value for which the portion of considered surface is illuminated or in shadow. Such horizon, expressed as a list of values, can be manually entered into the PVGIS interface or inserted as a parameter within the PVGIS API request in order to calculate the annual irradiance, averaged over a period of 10 or 12 years according to, respectively, the CMSAF and SARAH datasets. However, this procedure is extremely slow due to (i) the time required by GRASS GIS to calculate the horizon for each point and (ii) the number of API calls to the PVGIS remote service and the overall response time resulting from such communications.

This approach represents, however, a valid and already validated benchmark technique that allows for verifying the accuracy of the results obtained with the proposed methodology.

## 4. Results

As a point example of application of the proposed approach, we consider the case of a building in the city center that, even if located in front of a square, is also surrounded by taller buildings that determine shading effects, as shown in Figure 3.

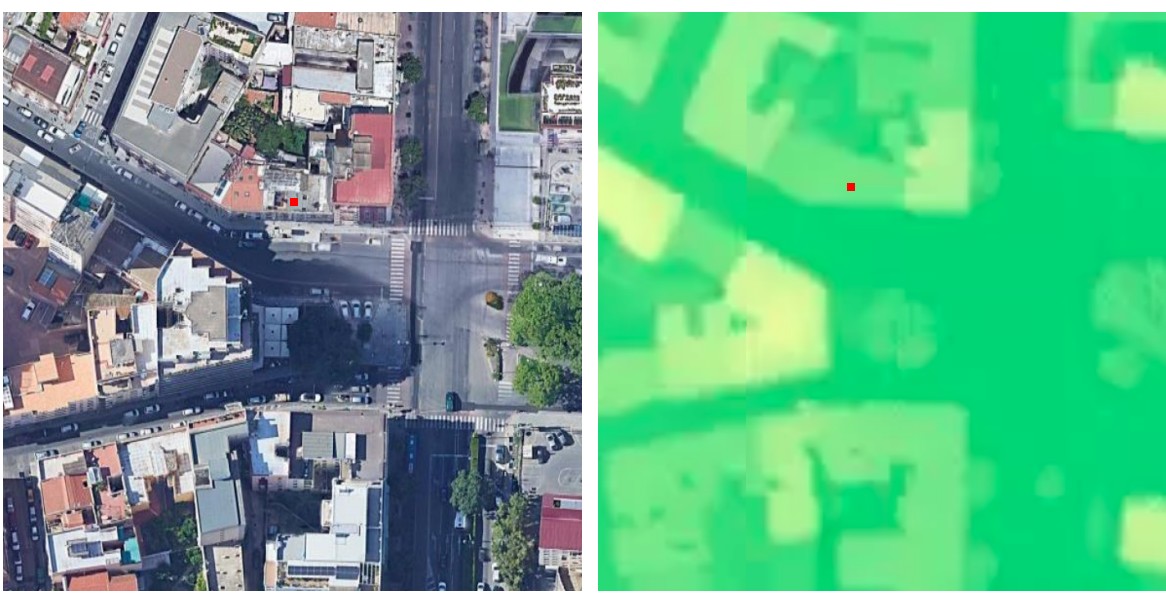

(**a**) Aerial view. (**b**) Elevation map.

**Figure 3.** Satellite view and elevation map of the building chosen as an example and the surrounding area. The roof of the considered building (identified by the red point) is at a lower elevation than the buildings on its left and right, and also with respect to the building located on the other side of the street in the south-west direction.

As described in Section 3.5, the irradiation time series for the area in question are first obtained through the PVGIS API, then the values of the irradiance components are grouped by discretized azimuth and zenith values, and are finally added up; the result is shown in Figure 4. For each apparent position of the Sun for which the irradiation components are not null, it is, therefore, possible to evaluate whether or not the area under examination is in the shade; Figure 5 shows the effect of shading on GHI as a function of the apparent position of the Sun for a typical year. It appears evident the reduction of radiation due to the effect of the shadows brought by the buildings.

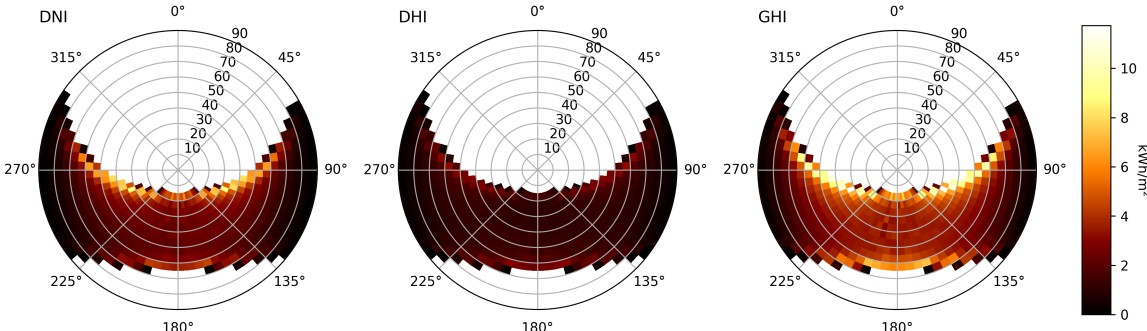

**Figure 4.** DNI (**left**), DHI (**center**) and GHI (**right**) irradiation for a unitary horizontal surface distributed over the celestial dome according to the apparent position of the Sun. The average annual irradiance for the macro-area of the building under examination, according to the PVGIS-CMSAF dataset, is estimated to 1819 kWh/m$^2$/year.

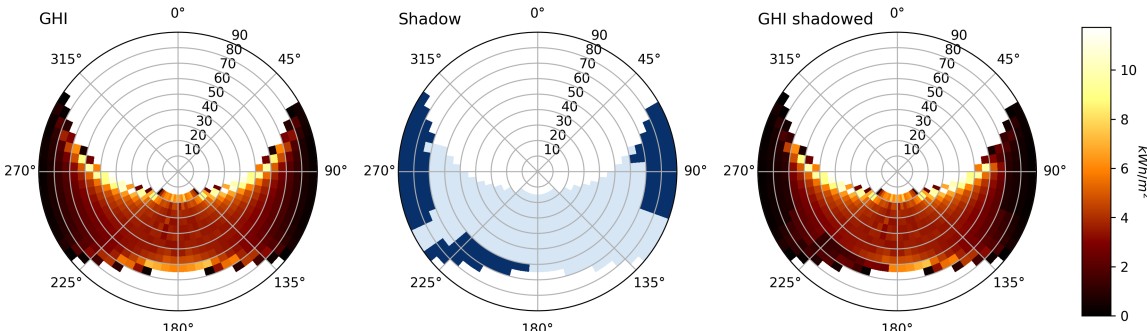

**Figure 5.** The left figure shows the GHI computed without any obstruction. The dark areas in the central figure show the shading for the selected location, i.e., the positions of the Sun for which the spot is in the shadow of nearby buildings. The plot on the right shows the actual GHI, computed by taking into account the shading. The actual annual irradiance, according to the PVGIS-CMSAF dataset, is 1647 kWh/m$^2$/year, i.e., corresponding to a 9.5% reduction with respect to the case without shading.

The estimation of the average monthly irradiance on a horizontal surface is shown in Figure 6 for the proposed approach and for the benchmark methodology described in Section 3.6. Such quantities are calculated by grouping and averaging the values of the time series output by PVGIS for both the CMSAF and SARAH datasets. The graphs show the monthly average values and the lower and upper extremes during the 10 and 12 years, respectively, for which the time series where available. The latter demonstrate the variability of the irradiation for each month across the years. The estimated GHI for the two methodologies substantially match with minimal deviations. For the considered location the two techniques provide an estimate of the annual irradiance on a horizontal surface equal to 1647 kWh/m$^2$/year for the proposed method and to 1660 kWh/m$^2$/year for the reference method, with a relative deviation equal to 0.8% (CMSAF), and respectively of 1602 kWh/m$^2$/year and 1617 kWh/m$^2$/year, with a difference of 0.9%, for the SARAH dataset.

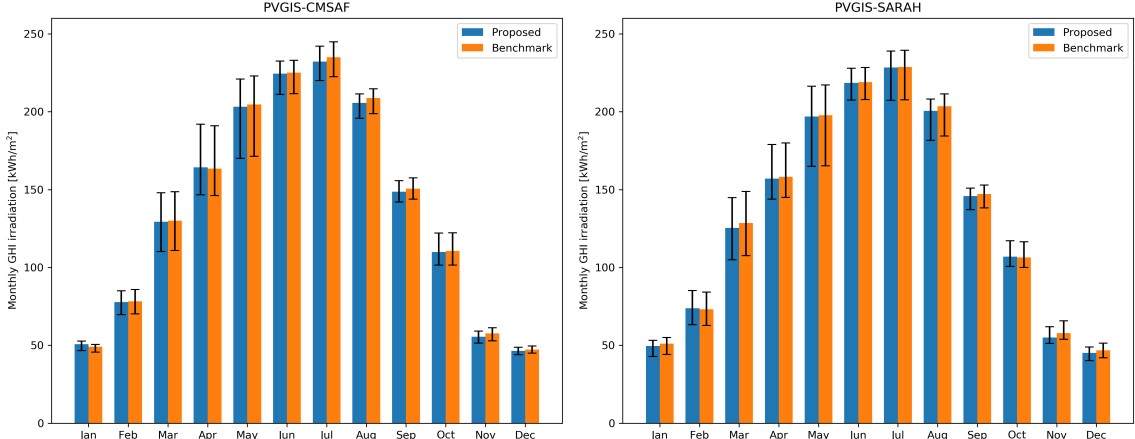

**Figure 6.** Comparison of the GHI radiation calculated for the building under examination according to the proposed methodology and to the reference approach for both the PVGIS-CMSAF (**left**) and the PVGIS-SARAH (**right**) datasets. The bars correspond to the mean value for the datasets; the interval of possible values of monthly average over the 10 (CMSAF) and 12 (SARAH) years in the dataset is also shown.

The entire city domain, which includes almost 9000 buildings for a theoretically available roof area of $4.68 \times 10^6$ m$^2$, was then processed with the proposed methodology. After filtering with the procedure described in Section 2.2, the useful surface has been reduced by 21% to $3.7 \times 10^6$ m$^2$, i.e., the 79% of the total rooftops.

With reference to the CMSAF dataset on the filtered surface, an average annual irradiance on horizontal surface of 1736 kWh/m$^2$/year is estimated considering the shading effect, while, not taking shading into account, a nominal value of 1842 kWh/m$^2$/year is calculated, thus indicating a loss of 5.75% due to shading. According to our estimate, therefore, the total irradiation on a horizontal surface for all the rooftop surfaces suitable for the installation of photovoltaic panels and within the city of Cagliari is equal to about 6.42 TWh/year. The corresponding values for the SARAH dataset are 1659 kWh/m$^2$/year for the average irradiance on horizontal surface and 1747 kWh/m$^2$/year when the effect of shading is not accounted for, a reduction due to shading equal to 5.03% and 6.13 TWh/year of total average annual irradiation for surfaces suitable for installation of photovoltaic panels.

The average error between the two approaches is quantified, in the case of the CMSAF dataset, with a mean absolute error (MAE) of 28.25 kWh/m$^2$/year and a root mean squared error (RMSE) of 60.72 kWh/m$^2$/year for an average GHI of 1736 kWh/m$^2$/year. The MAE is 1.6% and the RMSE is the 3.5% of the average value. When using the SARAH model data, we have a better match between the estimates: MAE = 22.95 kWh/m$^2$/year (1.4%) and RMSE = 51.47 kWh/m$^2$/year (3.1%), with GHI averaging 1659 kWh/m$^2$/year.

Figure 7 shows the monthly horizontal surface irradiation averaged over the entire domain for both the CMSAF and the SARAH datasets; in this case too there is a substantial match of the GHI estimates for the proposed methodology and the reference technique. The deviation between the two methods is minimal and mainly limited to the winter months when the CMSAF data are used and to the summer months when the SARAH dataset is employed.

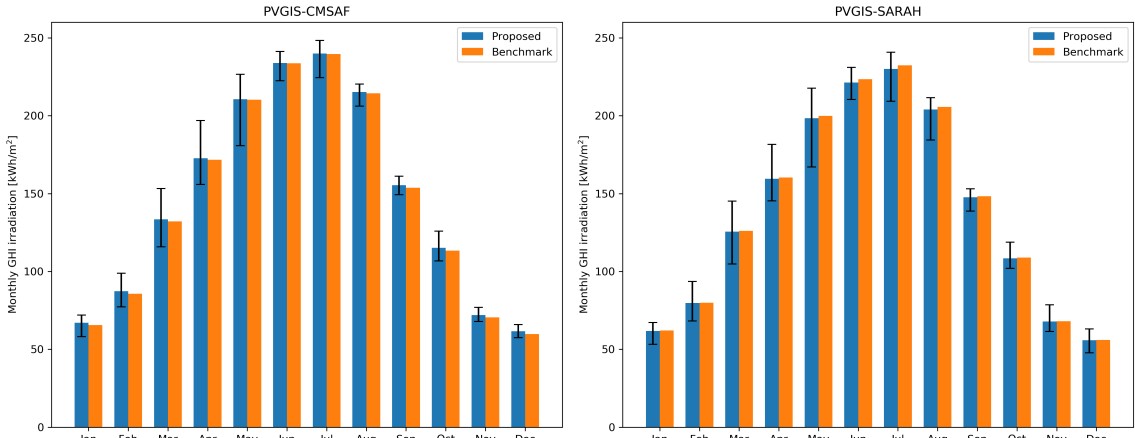

**Figure 7.** Comparison of GHI radiation calculated for the whole city of Cagliari according to the proposed methodology and to the comparison approach for the PVGIS-CMSAF dataset (**left**) and the PVGIS-SARAH dataset (**right**).

Figure 8 shows a detail of the GHI distribution over the building rooftops: shading might result in significant reductions in irradiation. The lower values of GHI are obtained for buildings surrounded by taller buildings, but in general the GHI radiation values appear distributed uniformly and close to the maximum value obtainable for a free surface.

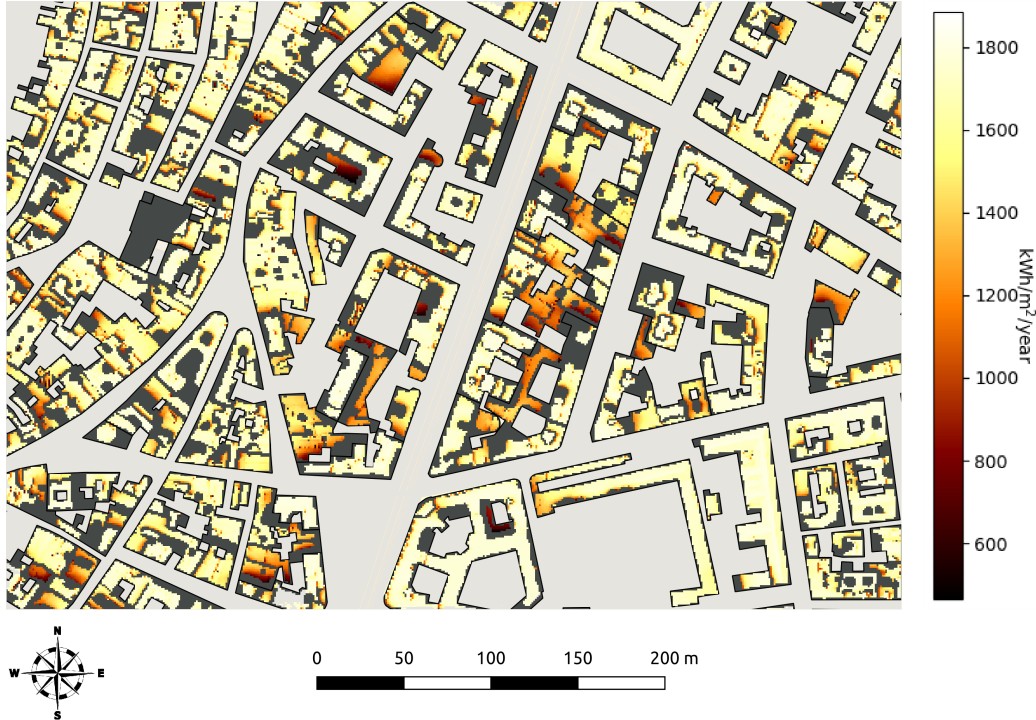

**Figure 8.** Annual GHI values (in kWh/m$^2$/year), calculated according to the PVGIS-CMSAF dataset, for cells of 1 m$^2$ belonging to contiguous surfaces with an extension of at least $S_{min}$ = 30 m$^2$ and with a slope of less than $\beta_{max}$ = 45°, in a district of Cagliari. Roof portions that do not meet the criteria are shown in dark grey.

The accurate estimation of the irradiation on the surface of the panels is necessary for evaluating the potential generation from a photovoltaic system. In this approach we assume that panels are always built-in with the rooftop surfaces, i.e., respecting the slope of the surface on which they lie

on, and therefore will be characterized by the tilt and the orientation of the rooftop on which they are mounted.

Unfavorably oriented surfaces such as north-facing rooftops, have a meager profitability due to the low average annual irradiance received (Figure 9).

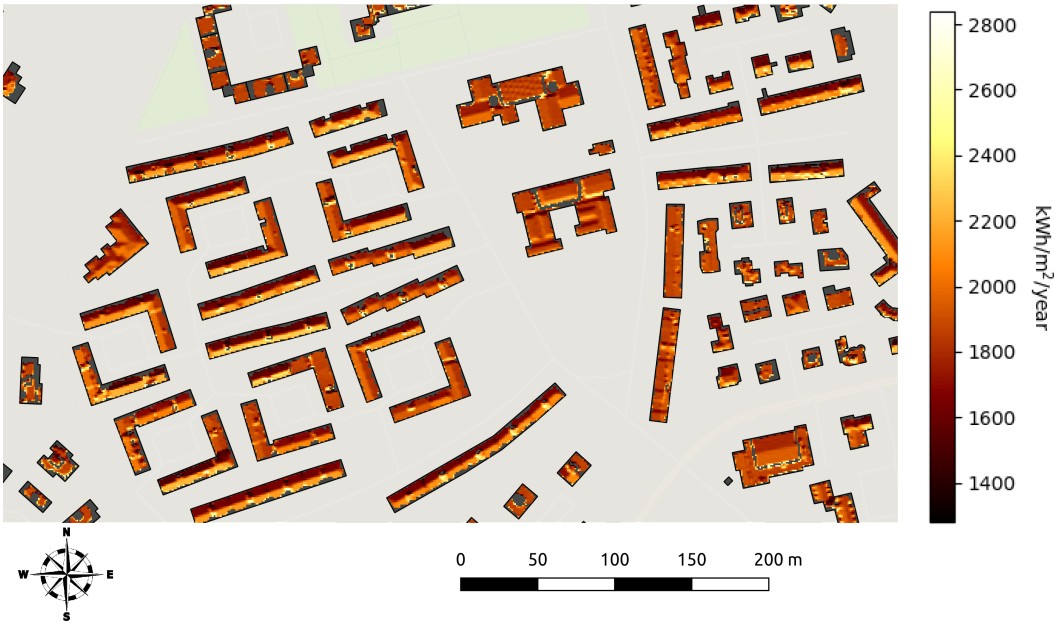

**Figure 9.** Annual values of the estimated POA irradiation (in kWh/m$^2$/year, according to the CMSAF dataset) produced by photovoltaic systems installed on suitable rooftop surfaces. The productive advantage of the south-oriented panels is glaring; moreover, the advantage given by panels with sub-optimal inclination, with the same southern orientation, can still be appreciated.

The total irradiance on the plane of the panels for the municipality of Cagliari, considering the inclination of the panels themselves in addition to the effect of shadows between buildings has an annual technical potential equal to approximately 6.89 TWh/year according to the irradiation provided by the CMSAF dataset and of about 6.54 TWh/year when processing the measurements provided by the SARAH dataset. The electric power potential can be estimated assuming a conversion efficiency for the PV panels.

The left plot in Figure 10 shows the distribution of GHI values calculated according to the proposed method for the whole set of filtered rooftop surfaces with a resolution of 1 m$^2$. The distribution is characterized by the presence of different peaks: this is in fact the result of the composition of the GHI distributions for each macro-area (approximately 80% of the surface area of the buildings is actually distributed over just 5 macro-areas, out of a total of 32, in the case of the PVGIS-CMSAF dataset: the buildings within the city of Cagliari are in fact grouped in a quite small area compared to the whole municipal territory.). The most evident peaks correspond to the macro-areas with the highest number of buildings, i.e., the most surfaces potentially usable for the installation of photovoltaic panels. These differences are due to the solar irradiation data of the PVGIS datasets: the values can be significantly different as the result of distinct average weather conditions due, for example, to the proximity to the sea or another body of water, or the presence of a high grounds within the macro-area. The irradiance on the POA surfaces is characterized by the distribution shown in Figure 10, where a larger ratio of building rooftops suffer from a reduction in actual irradiance, compared to the result for GHI. The effect is mainly related to the panels facing north which collect less radiation. However, the effect is mitigated by the more favorably oriented tilted roofs, which enjoy not only a superior POA irradiation, but also a larger exposed area due to the rooftops inclination.

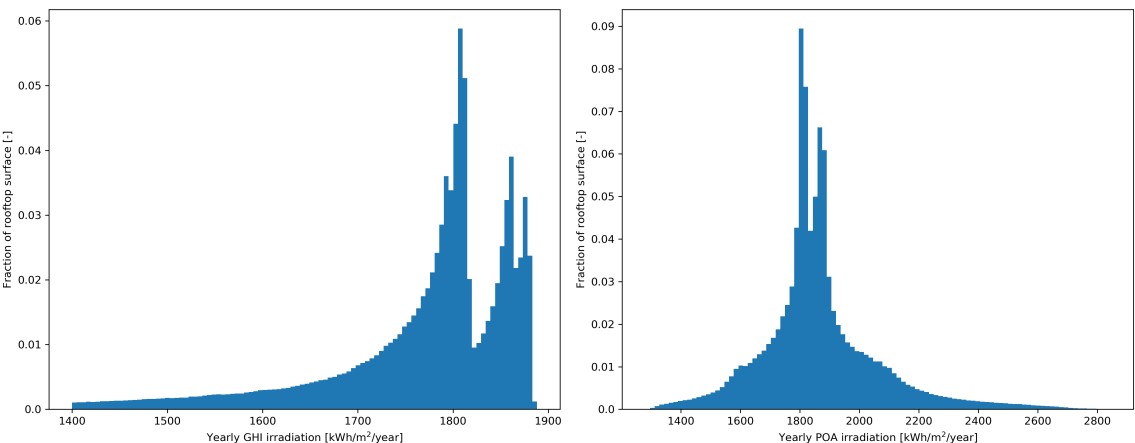

**Figure 10.** Distribution of the suitable rooftop surface fraction as a function of the GHI (**left**) and POA irradiation (**right**). For a clearer view, the left tail of the GHI right-modal distribution has been truncated at GHI = 1400 kWh/m$^2$/year.

The assessment of the roof surface area available for installation is based, as mentioned, on two threshold values. The first is related to the value of the slope computed from the DSM, beyond which the portion of the surface in question is considered unsuitable for the installation of panels, because at that point there are technical obstacles due, for example, to on-roof installations. A threshold of 45° was chosen. The second is represented by the value of the minimum usable surface area for a given building, below which the installation of a system is not technically justified. A threshold of 30 m$^2$ was set. We verified the sensitivity of the estimates obtained for GHI and POA irradiation to the variation of these threshold values, the results are shown in Figure 11. From the values shown it can be seen that as the threshold value for the minimum available area increases, the constraint becomes more stringent and the smaller, unattractive areas are excluded from the total count. On average there is a reduction in the total irradiation both for GHI and POA, while the average value on the surfaces is almost unchanged. As the threshold on the slope increases, the total area considered acceptable increases, as the constraint becomes less stringent, and consequently the total irradiation collected increases. Also in this case the average values of the GHI and the POA irradiations on the resulting suitable surfaces do not show significant changes.

In addition to assessing the technical feasibility of installing a photovoltaic panel on a given surface, it would also be necessary to consider the cost-effectiveness of such installations. Since, with the same surface area of the panel, its profitability is directly proportional to the POA irradiation it can collect, it is possible to think of a threshold on the POA irradiation above which it is economically convenient to install the panel. Figure 12 shows the fraction of roof surfaces that exceed this economic threshold as a function of the average irradiance collected. Both the fraction of the total surfaces and the portion exceeding the technical constraints (equal to 79% of the total) are shown. By imposing an additional constraint on the minimum allowable irradiance, the total potential can clearly be reduced. An in-depth economic-commercial analysis is beyond the scope of this work.

Summarizing, the proposed procedure is fast when compared to the benchmark technique, with calculation times for both methods reported in Table 1. The calculation of horizons with GRASS GIS is computationally expensive and, in order to be tackled in an acceptable timeframe, requires the parallelization of the process on a distributed cluster; on the contrary, the proposed methodology can be completed in about half an hour on a low-performance laptop. For the benchmark methodology it is also necessary to use a database to store the large amount of downloaded and processed data; in the proposed methodology the whole procedure can instead be completed without the need to store on disk the irradiation time series and the arrays containing the heights of the horizons.

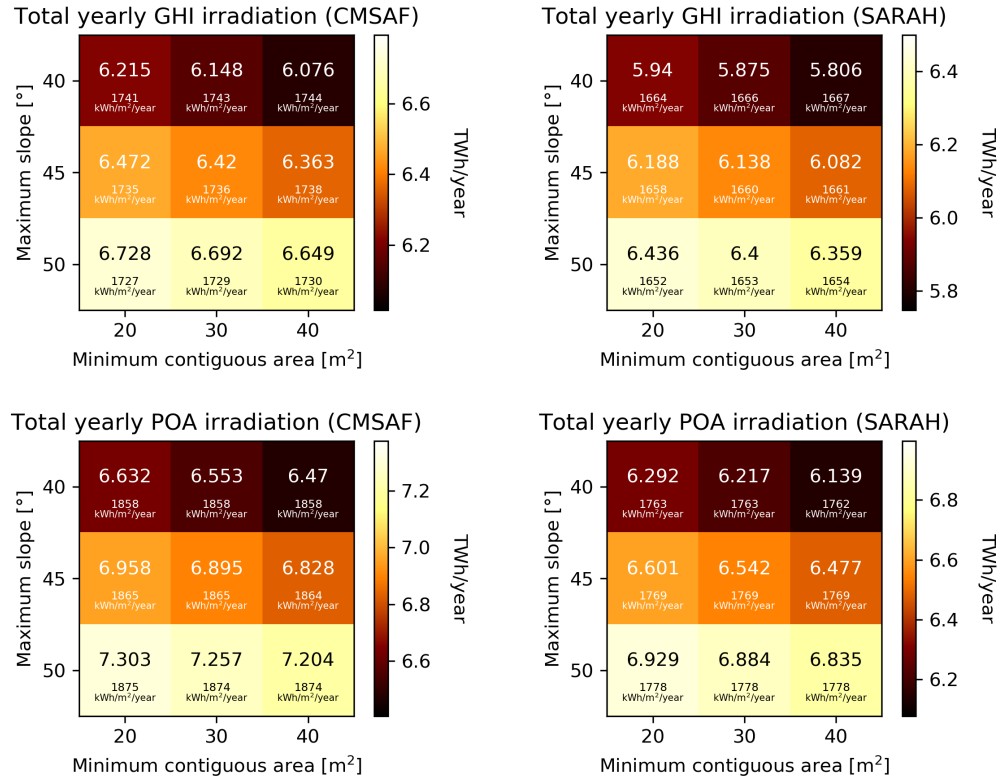

**Figure 11.** Variation of the total GHI (**top**) and POA (**bottom**) irradiance collected from the totality of suitable surfaces, as the threshold parameters for the minimum installation surface and the maximum slope angle change, for the two radiation datasets CMSAF (**left**) and SARAH (**right**).

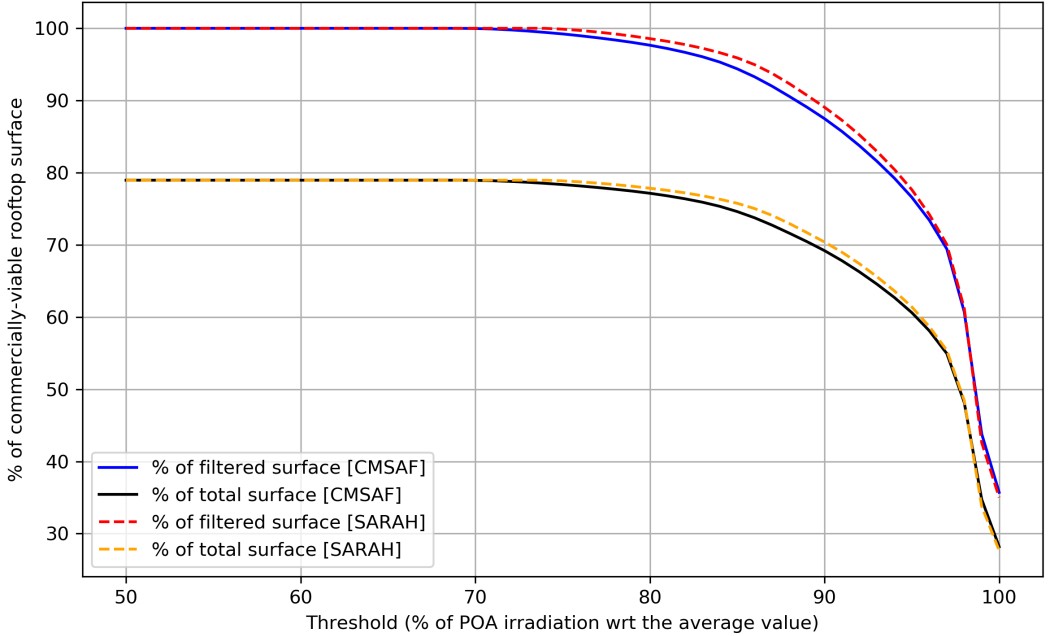

**Figure 12.** Variation in the percentage of usable rooftop surfaces with respect to the fraction of the average POA irradiation for which installing a photovoltaic system is deemed economically profitable.

**Table 1.** Processing time required to perform the proposed and the reference procedures on a single i7-4820K @ 3.70 GHz core for the 3.7 million 1 m$^2$ cells which compose the building rooftops. For the calculation of horizons using the GRASS GIS `r.horizon` functionality, a computer cluster was actually used with the aim of distributing the processes and thus reducing the calculation time to about 1 week.

| Methodology | Proposed | Benchmark |
|---|---|---|
| Horizon computation | (not applicable) | ≈1 year |
| Shading computation | ≈20 min | (included in the next step) |
| PVGIS service | <1 min (10–20 API requests) | ≈ 2–3 weeks (3.7 millions API requests) |
| Data processing | ≈5 min | (included in the previous step) |
| **Total** | <30 min | >1 year |

## 5. Discussion

The proposed methodology produces reliable results when compared with a state-of-the-art and widely used technique, and furthermore allows for a rapid and effective assessment of the complete census photovoltaic potential in urban areas.

The procedure has been applied to the case of the city of Cagliari, which is of particular interest for the projects in which this work is included. A direct comparison of the results with those obtained in other cities is clearly difficult, due to the differences in the type of buildings, the geographical position and the meteorological peculiarities of the site. The most directly comparable data is that relating to the rooftop surface actually exploitable.

The production potential for the city of Seoul was analyzed in [7], where the estimation of the rooftop surface available for the installation of photovoltaic modules was based on cadastral data and usage coefficients; moreover, the effect of the panel inclination on a flat surface was evaluated. The authors estimated a surface actually available for the installation of photovoltaic panels equal to about 38% of the available surface in case of zero slope and equal to 21% for an inclination of 20° corresponding to the optimal inclination for an isolated installation.

In [12], the generation potential for the Gangnam District of Seoul is evaluated including the effects of shading and estimating an overall availability of rooftop surfaces suitable for the installation of photovoltaic panels equal to the 66% of the total rooftop areas.

In [14] the production potential for several cities in the United States is evaluated by including the effects of shading and roof orientation: the 32% of the total building rooftop area is estimated as suitable for the installation of photovoltaic modules.

In the case of the recent Europe-wide study in [5], the radiation processing was carried out on a 100 m resolution raster grid in order to evaluate the technical and economic potential of photovoltaic production for the whole European territory. In this work, the portion of rooftop surface actually available for the installation of panels is estimated to be 60% of the total.

According to our calculations for the city of Cagliari, we would obtain a surface area suitable for the installation of photovoltaic panels equal to about 60% of the total surface area of the building rooftops, in line with the results available in the literature, if it would be economically advantageous to settle for a production equal to 95% of the average value of production for that area. In reality, the suitable surface area should be equal to almost 80% of the total rooftop area if it were sufficiently profitable to have a production equal to about 80% of the average production, as shown in Figure 12.

## 6. Conclusions

Although several tools are available for the estimation of the production of a single photovoltaic system, even in the presence of shading due to fixed obstacles, these are not computationally effective when applied in large urban areas to obtain an high resolution estimate of the production potential.

In this work, we proposed a procedure to estimate the generation potential in urban areas quickly and effectively by introducing a technique for aggregating radiation data and an algorithm for the fast

calculation of shadows cast by surrounding buildings and vegetation. We applied the method to the whole the city of Cagliari, obtaining an estimate of the yearly production potential. The results are practically coincident with those calculated by applying a reference methodology, but the proposed method requires only a fraction of the computational times and resources.

The developed technique is also suitable for application in other contexts, provided that high resolution data are available with respect to the elevation map and the description of the rooftop surfaces. Future developments of this work will be in the study of the compatibility of photovoltaic generation with city consumption profiles, with particular reference to the coupling with storage systems associated with electric mobility.

**Author Contributions:** The authors (A.P. and L.M.) jointly conceived of and designed the methodologies, performed the analysis and wrote the paper. All authors have read and agreed to the published version of the manuscript.

**Funding:** This work was partially funded by the Sardinian Regional Authorities with the projects "SARDINE" and "Tessuto Digitale Metropolitano", POR FESR Sardegna 2014–2020 Azione 1.2.2.

**Conflicts of Interest:** The authors declare no conflict of interest.

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
