# Peer review of "A Procedure for Complete Census Estimation of Rooftop Photovoltaic Potential in Urban Areas"

_smartcities, doi:10.3390/smartcities3030045_

Round 1

Reviewer 1 Report

The authors propose a methodology that allows a fast evaluation with minimal computational resources, and they apply the method to a city context and validate the results by comparison with a reference procedure.

The article is very interesting and this work has a direct application to other areas of the world.
The work is interesting for the readers of the magazine. The work is well written, and is novel and presented with quality.

However, I suggest in the introduction to use some current reference to photovoltaics.

Author Response

The authors propose a methodology that allows a fast evaluation with minimal computational resources, and they apply the method to a city context and validate the results by comparison with a reference procedure.

The article is very interesting and this work has a direct application to other areas of the world.The work is interesting for the readers of the magazine. The work is well written, and is novel and presented with quality.

However, I suggest in the introduction to use some current reference to photovoltaics.

We thank you for your appreciation of the work done, and we thank you for the suggestion. In the new version we have revised the introduction by adding additional reference to recent work on the subject. The changes are highlighted in green.

Reviewer 2 Report

Dear authors,

The approach is impressive in finding an very efficient way to take into account city morphology. However, before publishing, some minor adjustments to the manuscript should be undertaken. These are listed below.

Very minor grammatical errors. See attached file with cases which were identified (pink shading).

I do not agree with referencing people’s work by numbers. Prefer “J Doe et al. [x]”. I’ll leave this point to the editors.

Superscripts missing from many units in figures.

Abstract

Too short. Must include main methods, and quantify results and conclusions.

Introduction

The authors state that a significant fraction of PV must come from roof mounted systems. However, they do not give a good argument, eg when compared to ground mounted utility scale PV.

Methodology

Line 122: 45deg angle appears arbitrary. Authors should highlight this and discuss the potential impact of changing this value.

30m2 Lower limit is also arbitrary. However, authors do indicate that this is not a financial analysis paper. Would results be very different if this arbitrary value were set to eg 20m2 ou 40m2?

Line 139: Which physical model?

Line 189: Are the authors arguing that when a point is in shade they will not calculate the diffuse components of the radiation?

Method application

Figure 2: Clarity missing from figures. Which way is North? What is the red square explicitly? Figure 2b is not colour coded. Eg, what are the blue lines? What is the scale?

Figure 5: Colour code missing for “shadow”. Polar units from one graph superimpose on the adjacent graph.

Figure 6: What are the error bars?

Line 308: Maybe clarify and state that the irradiation is annual 6.42 kWh/yr

Figure 7: Colour coding is not perfectly clear/evident.

Figure 8: What are the brown bars?

Figure 10: I can only see two (and not several) peaks centred around 1800-1900 kWh/m2/yr

Figure 12: Why for the data produced from the SARAH database, does the % never go above 80% even for low thresholds?

Line 393: Sentence starting on this line is not clear.

-end-

Author Response

Dear authors,

The approach is impressive in finding an very efficient way to take into account city morphology. However, before publishing, some minor adjustments to the manuscript should be undertaken. These are listed below.

We thank you very much for your appreciation of the work and for the very useful comments you made, which we think have allowed us to improve the paper in the new revision. In the following, we provide a detailed response to your notes. We have revised the manuscript accordingly and you will find the corrections highlighted in yellow in the new version.

Very minor grammatical errors. See attached file with cases which were identified (pink shading).

Thank you very much for your help, we implemented all the suggestions of the notes in the accompanying pdf.

I do not agree with referencing people’s work by numbers. Prefer “J Doe et al. ”. I’ll leave this point to the editors.

We agree, we have modified the text to include explicit reference to the authors whenever possible, still following the style of the journal.

Superscripts missing from many units in figures.

You are absolutely right, we have modified the figures with an improved unit formatting.

Abstract

Too short. Must include main methods, and quantify results and conclusions.

Thanks for pointing it out. We have extended the abstract as you suggested, so as to have a fuller picture of the content of the work.

Introduction

The authors state that a significant fraction of PV must come from roof mounted systems. However, they do not give a good argument, eg when compared to ground mounted utility scale PV.

We thank you very much for your remark, we now realize that in translating the text we have unintentionally reversed the meaning of what we were trying to say. The purpose of the work is in fact to propose a procedure that allows evaluating the portion of solar radiation that can actually be collected through roof-mounted PV systems, to allow a technical and economic evaluation also in comparison to large installations as you correctly suggested. We have amended the text accordingly.

Methodology

Line 122: 45deg angle appears arbitrary. Authors should highlight this and discuss the potential impact of changing this value.

30m2 Lower limit is also arbitrary. However, authors do indicate that this is not a financial analysis paper. Would results be very different if this arbitrary value were set to eg 20m2 ou 40m2?

We thank you for pointing that out to us. The two values are at least in part arbitrary and related to the typical conformation of buildings in the environment considered, where the roofs are flat or with reduced slopes, and where, as mentioned in the text, for the current costs and radiation characteristics, 30m2, are considered the minimum size for which the construction of a system can be justified. We have carried out the calculation for other values of these parameters, varying the threshold of the surface at 20m2 and 40m2 and varying the maximum slope to 40° and 50°, we reported the results to highlight the sensitivity to these parameters. We have included the results, a new figure, and the relative discussion in the new revision of the paper.

Line 139: Which physical model?

We thank you very much for your comment, which allowed us to notice an inconsistency in the text. Our calculations were limited to the estimation of the radiation on the panel plane, we did not extend the calculation to the estimation of the actual power due to the different technologies available, with different costs and different efficiencies, because we would have shifted the discussion to the aspect of economic feasibility. We have modified the text to better clarify this aspect and we have indicated some physical models that can be used for the estimation of electrical power.

Line 189: Are the authors arguing that when a point is in shade they will not calculate the diffuse components of the radiation?

We are sorry for the misunderstanding. The diffuse component is not masked by the shadow. We have modified the text in order to make it clearer.

Method application

Figure 2: Clarity missing from figures. Which way is North? What is the red square explicitly? Figure 2b is not colour coded. Eg, what are the blue lines? What is the scale?

Thank you for your useful observations. We have modified figures 2a and 2b, and updated the caption, inserting the unit of measurement for the color bar, indicating the north and the scales, explaining the meaning of the red square, and modified the map layer to put in evidence the roof surfaces only.

Figure 5: Colour code missing for “shadow”. Polar units from one graph superimpose on the adjacent graph.

Thank you very much for bringing this to our attention. We have redone the graphs. We modified the caption of the figure to clarify the color code used, we thought not to add a second axis to keep the same format of figure 4.

Figure 6: What are the error bars?

Thank you for your comment. These are the minimum and maximum values of the average monthly irradiation in the 10 years of dataset readings as briefly mentioned in the text. We have calculated the value for all 10 years and grouped it on the months, so as to highlight the possible variations from year to year, in order to give a reference to the relative importance of approximation in estimating irradiation. In the new revision, we have updated the text of the caption to clarify this.

Line 308: Maybe clarify and state that the irradiation is annual 6.42 kWh/yr

Thank you for bringing this to our attention, we have corrected the text as suggested

Figure 7: Colour coding is not perfectly clear/evident.

Thank you for your comment. We have modified the layer of the map, highlighting only the surfaces of the buildings, for greater clarity, in a similar way to what we did for figure 2b.

Figure 8: What are the brown bars?

We're sorry the figure wasn't clear enough. In Figure 8 we compared the distribution of the GHI estimates obtained with the PVGIS system with those obtained with our procedure, it was a histogram with the number of 1m2 cells as a function of the calculated value of GHI on said cell. We modified Figure 8 by merging it with Figure 10 and reporting the distribution of surface fraction subject to any given irradiation on the horizontal plane and on the plane itself. The ordinate represents the fraction of surface area subject to a given annual average irradiance with respect to the total. We have modified the caption and text accordingly.

Figure 10: I can only see two (and not several) peaks centred around 1800-1900 kWh/m2/yr

You are absolutely right, the resolution of the histogram allows to appreciate only two main peaks in the distribution. We have modified the text to better highlight this aspect and have grouped this plot with the previous distribution in figure 10 of the new revision.

Figure 12: Why for the data produced from the SARAH database, does the % never go above 80% even for low thresholds?

We thank you for this point, which allowed us to realize that we were not clear enough in the exposition. The 79% limit is relative to the portion of the total surfaces that is suitable for the installation of the panels. We have modified the figure, the caption, and the description of the results to better clarify this aspect.

Line 393: Sentence starting on this line is not clear.

We thank you for your comment. We have rewritten the whole paragraph, hoping that the new version will be clearer.

-end-

We thank you again for your very helpful comments and suggestions, we hope that the new version of the paper will be satisfactory.